# Untargeted Metabolomics Approach Using UHPLC-HRMS to Unravel the Impact of Fermentation on Color and Phenolic Composition of Rosé Wines

**DOI:** 10.3390/molecules28155748

**Published:** 2023-07-29

**Authors:** Cécile Leborgne, Emmanuelle Meudec, Nicolas Sommerer, Gilles Masson, Jean-Roch Mouret, Véronique Cheynier

**Affiliations:** 1UE PR, INRAE, Domaine de Pech Rouge, F-11430 Gruissan, France; 2SPO, INRAE, Univ de Montpellier, Institut Agro, F-34060 Montpellier, France; emmanuelle.meudec@inrae.fr (E.M.); nicolas.sommerer@inrae.fr (N.S.); jean-roch.mouret@inrae.fr (J.-R.M.); veronique.cheynier@inrae.fr (V.C.); 3INRAE, PROBE Research Infrastructure, Polyphenol Analytical Facility, F-34060 Montpellier, France; 4Institut Français de la Vigne et du Vin, Centre du Rosé, F-83550 Vidauban, France; gilles.masson@vignevin.com

**Keywords:** rosé wine, polyphenols, fermentation, color, untargeted analysis, UHPLC-HRMS

## Abstract

Color is a major quality trait of rosé wines due to their packaging in clear glass bottles. This color is due to the presence of phenolic pigments extracted from grapes to wines and products of reactions taking place during the wine-making process. This study focuses on changes occurring during the alcoholic fermentation of Syrah, Grenache and Cinsault musts, which were conducted at laboratory (250 mL) and pilot (100 L) scales. The color and phenolic composition of the musts and wines were analyzed using UV-visible spectrophotometry, and metabolomics fingerprints were acquired by ultra-high performance liquid chromatography−high-resolution mass spectrometry. Untargeted metabolomics data highlighted markers of fermentation stage (must or wine) and markers related to the grape variety (e.g., anthocyanins in Syrah, hydroxycinnamates and tryptophan derivatives in Grenache, norisoprenoids released during fermentation in Cinsault). Cinsault wines contained higher molecular weight compounds possibly resulting from the oxidation of phenolics, which may contribute to their high absorbance values.

## 1. Introduction

Rosé wine is much appreciated, and its worldwide production and consumption have increased by 20% in the last 20 years. Color is essential for the quality of rosé wines and a major driver of consumers’ purchase as they are usually sold in clear glass bottles [1]. Rosé wines are extremely diverse, covering a wide range of intensities and nuances, reflecting large differences in pigment compositions [2]. Indeed, rosé wines pigments contain phenolic compounds [2,3], including red anthocyanin pigments extracted from grapes and derived pigments formed during wine making through reactions of anthocyanins with other phenolic compounds and with wine metabolites such as acetaldehyde and pyruvic acid [4]. In addition, the rosé wine-making process, involving pressing before fermentation, i.e., when grape polyphenoloxidase is highly active, enables the enzymatic oxidation of hydroxycinnamic acids and flavan-3-ols, potentially leading to browning [5].

An investigation carried out on a large set of commercial rosé wines using UV-visible spectrophotometry, CIELab colorimetry and ultra-high-performance liquid chromatography coupled to triple quadrupole mass spectrometry (UHPLC-QqQ-MS) in the Multiple Reaction Monitoring (MRM) mode for the targeted metabolomics analysis of 125 phenolic compounds [6] has shown that the different color styles are due to different phenolic compositions [2]. Thus, the composition of light rosé wines was closer to that of white wines, with high proportions of hydroxycinnamic acids compared to anthocyanins, and consequently high levels of derived pigments resulting from reactions of anthocyanins with hydroxycinnamates (i.e., phenylpyranoanthocyanins) and with pyruvic acid, a yeast metabolite (i.e., carboxypyranoanthocyanins), explaining the red component of color a*. In contrast, darker rosé wines contained higher levels of anthocyanins and flavanols and pigments derived from these two families. The yellow component of color, b*, appeared mainly associated with compounds such as dihydroflavonols, aromatic amino acids, flavanols and anthocyanin–flavanol adducts, which might be involved in browning. However, the yellow pigments and most of the red pigments derived from anthocyanins and flavanols were not among the targets of the MRM method used.

The same methodology was used in combination with high performance size-exclusion chromatography coupled to UV-visible spectrophotometry to investigate color and phenolic composition changes occurring during the alcoholic fermentation of rosé musts [7]. This was performed on Grenache, Cinsault, and Syrah grapes, which are commonly used for the production of Provence rosé wine and selected for their different color potential [8]. A comparison of the data obtained using the three methods confirmed that a large part of the phenolic compounds measured by spectrophotometry in the Grenache and Cinsault musts were not detected by the MRM method and showed that they actually corresponded to larger oligomers and polymers likely derived from hydroxycinnamic acids and flavanols. The more intense red color of Syrah must was mostly due to anthocyanins which appeared more easily extracted from this variety despite the shorter maceration time (2 h, 8 h, and 12 h, respectively, for Syrah, Grenache, and Cinsault). A comparison of the must and wine compositions indicated losses of the oligomeric pigments during the fermentation of Cinsault and Grenache musts, which is mostly due to adsorption on lees. During the fermentation of Syrah musts, anthocyanins were partly converted to derived pigments, and only a limited color drop was observed. Thus, the must composition, resulting from grape composition but also from the extraction and oxidation of phenolic compounds during must preparation, impacted the fate of phenolic compounds during fermentation and consequently the wine composition and color. This may be due to varietal differences. Indeed, the higher content and extractability of anthocyanins have been reported for Syrah grapes [9]. However, differences in the duration of maceration and the higher levels of sulfur dioxide in the Syrah musts made it difficult to assess varietal specificities. Nevertheless, high-performance size-exclusion chromatography (HPSEC) analysis confirmed the presence of higher molecular weight phenolic compounds in the samples, especially in Grenache and Cinsault musts, that were missing in the targeted MRM method used.

The purpose of the present study was to compare the compositions of Grenache, Syrah, and Cinsault musts and wines made under identical wine-making conditions to determine whether the differences observed earlier were attributable to varietal characteristics or to process. The total phenol and pigment composition was evaluated by UV-visible spectrophotometry. Must and wine fingerprints were generated by untargeted chemometrics analysis based on ultra-high-performance liquid chromatography coupled to high-resolution mass spectrometry and processed with chemometrics approaches to detect clusters and markers that differentiate samples [10,11,12].

This was performed using untargeted metabolomics based on ultra-high-performance liquid chromatography coupled to high-resolution mass spectrometry in combination with UV-visible spectrophotometry. Laboratory-scale (250 mL) and pilot-scale (100 L) fermentation were also compared in this study.

## 2. Results and Discussion

### 2.1. Spectrophotometric Analysis

The CIELab space color parameters L* (lightness), a* (red/blue) and b* (yellow/green), reported in Table 1, showed qualitative and quantitative differences between all modalities. The three musts studied exhibited color differences linked to the grape characteristics since all pre-fermentation treatments were common. The Syrah must was the most colored with the highest value of a* and the lowest value of b*, resulting in a reddish hue. Grenache and Cinsault musts were lighter, with b* values slightly higher than a*, giving an orange hue to the musts. Color intensity was also higher in Cinsault than in Grenache (significantly higher a* and b* values, lower L* value). Wine CIELab parameters showed qualitative and quantitative differences similar to those observed at the must stage between the three grape varieties. Nevertheless, wines were lighter (higher L*, lower a* and b*) and slightly more orange (higher hue value) compared to the corresponding must. An impact of fermentation scale was also observed for some parameters. Grenache and Syrah wines fermented at laboratory scale (LSW) exhibited lower L* compared to wines fermented at pilot scale (PSW). Moreover, Cinsault LSW exhibited lower a* and Grenache LSW higher b* than the corresponding PSW. These data suggest a higher extent of oxidation in LSW, resulting in an increased degradation of red pigments combined with the formation of yellow pigments.

A previous study [7] showed the same color differences between musts and wines from Grenache, Cinsault, and Syrah, although maceration times were different, suggesting that these differences primarily reflect varietal differences in terms of phenolic composition and extractability.

#### UV-Visible Absorbance Characteristics

The total phenol index (TPI), total red pigments (TRP) and absorbance at 320 nm (A_320_) presented in Figure 1 showed different intensities and ratios, indicating different phenolic profiles between the three grape varieties along with changes induced by alcoholic fermentation.

Must TRP values were significantly different, Syrah must showing the highest value and Grenache the lowest, which is consistent with color parameters. Syrah must also exhibited a significantly higher TPI value than the other musts but an intermediate value of A_320_. Grenache must showed a significantly higher value of A_320_ compared to other musts and an intermediate value of TPI, while Cinsault must had the lowest TPI and A_320_ values. TRP represented 30% of TPI in Syrah must, 13% in Cinsault and only 5% in Grenache. This comparison indicates the higher proportion of red anthocyanin pigments compared to other grape phenolics in the Syrah must. A_320_ values represented 120% of TPI in Grenache must, suggesting the presence of large quantities of hydroxycinnamic acid derivatives which show characteristic absorbance at this wavelength, as observed in white wines [13]. The contribution of hydroxycinnamic acid moieties to the total phenol content in the Cinsault must (80%) was lower than in the Grenache must but higher than in the Syrah must (60%).

Differences in the relative intensities and ratios of TPI, A_320_, and TRP between wines made from the three grape varieties were similar to those described for the three corresponding musts, although all intensities were lower. Indeed, Syrah wines demonstrated the highest TPI and TRP values along with an intermediate absorbance value at 320 nm compared to the other wines. Grenache wines showed higher absorbance at 320 nm, intermediate TPI and the lowest TRP values, as observed on the musts. Cinsault wines showed the lowest TPI and A_320_ values, and their TRP values were similar to those of Grenache wines. No significant difference was observed between the two fermentation scales, except for the Syrah TPI, A_320_, and TRP values and the Grenache A_320_ value, which were lower in the wines fermented at laboratory scale, again suggesting higher oxidation rate. In addition, the ratio of A_320_ to TPI remained stable from must to wines.

Losses during alcoholic fermentations at pilot scale were similar for Grenache and Cinsault (10–15% for TPI, 60–70% for TRP and 10–20% for A_320_) and lower for Syrah (6% for TPI, 20% for TRP and less than 1% for A_320_). For Grenache and Cinsault, the losses were similar at both fermentation scales. Much higher losses were observed for laboratory-scale Syrah wines (LSW), reaching 20–25% for TPI and A_320_, and 60% for TRP. Thus, for all three varieties, TRP losses represented 20 to 70% of the initial must values while TPI dropped by only about 10–25%, indicating the high impact of fermentation on red pigments especially.

These data indicate that the different TPI values reflect different total phenol contents but also different proportions of the three major classes of phenolic compounds which have different extinction coefficients at 280 nm [14]. Thus, the Syrah must and wines contain a higher proportion of anthocyanins and a lower proportion of hydroxycinnamic acids compared to the other grape varieties, while the reverse is true for the Grenache.

Similar differences in phenolic profiles have been observed between Syrah, Cinsault, and Grenache musts and wines obtained respectively after 2 h, 8 h, and 12 h of maceration in an earlier study [7], suggesting that these differences are related to varietal characteristics. Indeed, Grenache has been reported to contain higher levels of hydroxycinnamic acids than the other two varieties, especially when considering must composition [15]. Moreover, compared to other common varieties, Syrah grapes have been reported to contain high levels of anthocyanins, which are also highly extractible [9]. Extraction also impacts phenolic composition and color, as shown by the increase in absorbance values with the extent of pressing [7]. Accordingly, the TPI values of Cinsault musts higher than those of Grenache musts [7] reflected higher extraction rates due to longer maceration (12 h versus 2 h for Syrah and 8 h for Grenache, 4 h for all three musts in the present study). However, in spite of the shorter maceration (2 h) used in the earlier study, Syrah musts contained higher levels of total phenols and total red pigments than that described in the present work, which was likely due to differences in the grape composition or to the higher SO_2_ level, protecting phenolic compounds against oxidation during must preparation.

In wine and must, anthocyanins and their derivatives are present in various colored or uncolored forms [16]. Flavylium cation (AH^+^) and non-bleachable pigments (NBP) (i.e., pyranoanthocyanins) are colored in musts and wines, whereas hemiketal form (HF) and bisulfite adducts (BA) are colorless. The four different forms of red pigments in musts and wines of Grenache, Cinsault, and Syrah are presented in Figure 2. Like TRP, A_520_ values measured directly on musts and wines (A_520_) and after the addition of ethanal (A_520-ethanal_) along with hydrated forms (HF) and flavylium forms (AH^+^) of pigments calculated from the absorbance values were significantly higher in Syrah than in Cinsault, which was itself higher than Grenache. In addition, must values were significantly higher compared to corresponding wines.

Non-bleachable pigments (NBP) were significantly higher in Cinsault (0.17) than in Grenache (0.06) musts, while the Syrah (0.13) NBP quantity was intermediate. NBP accounted for 10–15% of TRP in Grenache and Cinsault musts, whereas they represented less than 5% of TRP in Syrah must. NBP quantity remained stable after fermentation for Grenache and Syrah, while a loss of 35% was measured for Cinsault. However, the proportion of NBP was higher than in the musts, accounting for 15–35% of TRP in Grenache and Cinsault wines and 5–9% in Syrah wines. A previous article reported similar proportions and values for Grenache and Cinsault musts. Syrah must showed very low amounts (0.06) of NBP compared to this study, which were explained by the higher concentration of sulfite (80 mg/L) and bisulfite adducts [7]. Bisulfite adducts (BA) were detected in very low amounts only in Grenache (0.04) and Syrah musts (0.06), representing, respectively, 7% and 2% of TRP. After fermentation, BA was detected in very low amounts mainly in wines fermented at pilot scale, confirming the higher oxidation rates in wines fermented at laboratory scale. Also, bisulfite adducts disappeared during alcoholic fermentation due to sulfite combination with the acetaldehyde produced by yeasts [17,18].

The decrease in TRP values and color drop during fermentation are due to losses of AH^+^ and HF forms of sulfite bleachable pigments. The proportions of AH^+^ and HF remained stable during the fermentation of Grenache (30/70) and Cinsault (45/55) at both fermentation scales, while the ratio of AH^+^ to HF decreased during fermentation of the Syrah must, from 30/70 to 25/75 (pilot scale) and 15/85 (laboratory scale). Except for Syrah wine fermented at laboratory scale, the proportions of flavylium cations exceed those calculated (12–15%) using the hydration constant of malvidin-3-*O*-glucoside (K_h_ = 10^−2.6^), pH values of musts and wines and pH formula calculation (pH = pK_h_ + log($\frac{HF}{AH^{+}}$) [19]. This might be due to the presence of pigments such as coumaroylated anthocyanins or ethyl-linked anthocyanin–flavanols [20,21], in which the flavylium moiety is protected from hydration by intramolecular or intermolecular copigmentation or self-association [22,23].

To conclude, differences related to the characteristics of the three studied varieties observed at the must stage were conserved in the wines, confirming the results of an earlier study [7]. Small differences between the two fermentation scales were also observed and attributed to enhanced oxidation at the laboratory scale.

### 2.2. Untargeted Metabolomics by High Resolution Mass Spectrometry

Differences between must and wine samples were further investigated by untargeted metabolomics analysis using UHPLC-HRMS. Therefore, all samples were randomly injected, in a single batch, along with QC and control samples, as described in metabolomics studies [11]. Positive ion mode was selected because it is better suited for the detection of anthocyanin pigments. The features obtained were filtered as described in the Materials and Methods section. The resulting 729 feature dataset was used for multivariate statistical analysis.

The PCA plot of the metabolomics spaces obtained by sample injections in the positive ion mode is shown in Figure 3.

The projections of QC samples, consisting of a mixture of all analyzed samples and regularly injected throughout the sequence, were very close together in a central position, indicating the reliability of the dataset. This unsupervised analysis also showed that the Cinsault must and wine samples, on the one hand, and the Syrah must, on the other hand, are separated from all other samples along the first principal component, while musts, pilot-scale wines, and laboratory-scale wines are distinguished on the second axis.

Molecular formula tentatively attributed from LC-HRMS signals by the Compound Discoverer software suggested that many of the metabolites belong to phenolic compounds but also to compounds containing nitrogen atoms such as amino acids, peptides, and indoles, as reported by earlier wine metabolomics studies [24,25]. Annotation was performed manually by comparing retention times, mass spectra and, when available, UV-visible spectra with an internal database and with data published in the literature [24,25,26,27]. A given molecule could be represented by several features with the same retention time but different *m*/*z* values, corresponding to molecular or pseudo-molecular ions, adduct and fragment ions, and some isotopic clusters. The heatmap (Figure 4) obtained by unsupervised hierarchical cluster analysis on the 729 selected features (Figure 3) confirmed that injections of triplicate biological samples are grouped together and showed separations between musts and wines and between varieties. However, the Grenache wine made at laboratory scale was the most different sample, while the pilot-scale Grenache wine clustered with the Grenache must. This analysis also highlighted clusters of features that were over- or underexpressed in some modalities. Clusters were defined as a group of features separated from three nodes on the hierarchical classification, as this enabled the discrimination of varieties and/or wine-making modalities. In addition, two-way ANOVA was performed (*p* < 0.05) for each feature to determine whether differences between two modalities were significant.

Thus, the first group (A—73 features), overexpressed in Syrah musts and wine samples, contains features corresponding to anthocyanins, flavonols, flavanol monomers and resveratrol. This is in agreement with the results obtained in an earlier study comparing the three varieties [7] and reflects the higher concentration of these phenolic compounds in Syrah musts and wines due to the higher grape content and/or higher extraction rate from Syrah grape berries.

A second cluster (B—126 features) contains compounds that appear specific to the laboratory-scale wines and overexpressed in Grenache ones and, for some of them, in the Syrah must. A large majority (118) were attributed to molecular formula containing nitrogen atoms, suggesting that they are amino acid derivatives. Moreover, the mean ratio of molecular weight to number of nitrogen atoms of cluster B features corresponds to the average molecular weight of amino acids (~110 g/mol), supporting their peptide annotation. Except for the subgroup also detected in Syrah must, originating from Syrah grapes, these compounds were thus formed during fermentation when conducted at laboratory scale, suggesting specific yeast metabolism or some microbial contamination.

The next cluster (C—103 features), specific to wines, contains both nitrogen compounds (46) and compounds that likely correspond to phenolic compounds. However, very few of these compounds could be identified: carboxypyranomalvidin-3-*O*-glucoside (i.e., vitisin A) formed by the reaction of malvidin-3-*O*-glucoside with pyruvic acid, a yeast metabolite released during fermentation, resveratrol, dihydroquercetin, and naringenin, likely arising from hydrolysis of their glycosylated derivatives during fermentation, *S. cerevisiae* metabolites such as panthothenic acid [28], tryptophan ethylester [29], and purine derivatives (guanine) and a number of unknown apolar compounds, potentially coumarin derivatives that may be degradation products of phenolic acids and/or flavonoids.

Cluster D (43 features) gathered compounds that were less abundant in Grenache than in Syrah and Cinsault but little affected by fermentation. Among them, a series of signals could be attributed to resveratrol dimers. These molecules are stilbenes that are synthesized by grapes, in particular as a defense mechanism against fungal attack [30]. Signals detected at 509.12912 and 347.0762 *m*/*z* were tentatively attributed to a rearrangement product of malvidin-3-*O-*glucoside and its aglycone fragment, as described earlier [31]. Other unknown compounds in this group might also correspond to phenolic derivatives.

Cluster E1 (93 features) corresponded to molecules that were more abundant in Cinsault than in the other two varieties but also more abundant in wines than in musts. This group contained in particular apolar compounds that were tentatively identified to norisoprenoid derivatives, that may be varietal markers, revealed during fermentation. Few phenolic compounds, including astilbin, a quercetin derivative, and ethylcoumarate were also tentatively identified in this cluster.

Compounds of cluster E2 (24 features) were more abundant in Cinsault musts and wines than in samples from the other varieties, and some of them were significantly more abundant in wines than in musts. This group contains higher molecular weight compounds, mostly containing nitrogen, detected as doubly charged ions. This particularity of Cinsault has already been observed in a previous study which revealed higher molecular weight compounds containing anthocyanin and hydrocycinnamic acid units as well as flavanols by size-exclusion chromatography coupled to UV-visible detection [7].

Cluster F (32 features) contained compounds that were significantly more abundant in Cinsault than in Grenache and in Grenache than in Syrah and, regardless of the variety, more abundant in musts than in wines. Most of these features (26) were attributed to molecular formula containing nitrogen atoms, suggesting that they are peptide derivatives from grapes, depending on the grape variety, and they are degraded during fermentation. Some of these signals differed by mass increments corresponding to one to three pentose moieties (C5H8O4; 132.042260 a.m.u), suggesting structures consisting of nitrogen compounds substituted with pentose chains.

Compounds gathered in cluster G (16 features) were also more abundant in musts but particularly in Grenache musts. The major signals identified correspond to tryptophan, whose content in grape depends on vine-growing conditions, especially nitrogen fertilization, and on ripening [32]. Other signals were tentatively attributed to products resulting from the enzymatic oxidation of phenolic compounds. Indeed, the singly charged ion detected at 928.15533 *m*/*z* can be attributed to a caftaric acid–GRP adduct, while the doubly charged ion at 617.11548 *m*/*z* (C_36_H_37_N_3_O_24_S) likely corresponds to a dimer of GRP (grape reaction product, i.e., 2-S-gutathionyl caftaric acid). GRP results from the nucleophilic addition of glutathione on the o-quinone generated by the enzymatic oxidation of caftaric acid. These adducts have never been reported in must or wine but are expected to form by the nucleophilic addition of GRP or caftaric acid onto GRP or caftaric acid o-quinone, resulting, respectively, from enzymatic oxidation and the coupled oxidation of GRP by the enzymically generated caftaric acid quinone [33]. The feature at 455.0611 *m*/*z* is a caftaric acid derivative as shown by the presence of a fragment ion at 323.05505 *m*/*z*, which is characteristic of the loss of a tartaric acid moiety (-C_4_H_4_O_5_: 132.00605 a.m.u). The latter structure may result from the oxidative coupling of caftaric acid and caffeic acid (or the hydrolysis of a tartaric ester bond of a caftaric acid dimer) followed by the loss of two water molecules. However, the loss of tartrate and water molecules may also occur through fragmentation in a mass spectrometry experiment, as shown earlier for similar products [34]. The mass signal at 575.11835 *m*/*z* (C_30_H_22_O_12_) likely corresponds to an oxidized flavanol dimer as described earlier [35]. The presence of these oxidation products in Grenache musts reflects the high level of hydroxycinnamic acids and especially caftaric acid, which is the major substrate for grape polyphenoloxidase, in this variety [5,15].

Compounds from cluster H (107 features) were over-represented in Syrah must. They comprised mostly nitrogen compounds, among which phenylalanine was identified. This cluster also included peonidin 3-glucoside and flavonol glycosides, again indicating higher berry contents and/or higher extraction rates for these compounds in Syrah grapes. A GRP isomer as well as GRP2, (2,5 di-S-glutathionyl caftaric acid) also belonged to this cluster. GRP and its isomer result from the reaction of glutathione with the o-quinone generated by the enzymatic oxidation of caftaric acid, while GRP2 is formed by the addition of glutathione on GRP o-quinone formed by the coupled oxidation of GRP by caftaric acid *o*-quinone [33]. The concentrations of these compounds depend on the proportion of glutathione to caftaric acid in the must, which is a varietal characteristic [5,15], and on the extent of enzymatic oxidation due to oxygen exposure during must preparation. However, higher concentrations in must than in wine suggest that these compounds undergo some degradation, possibly involving hydrolysis of the peptide bonds or further oxidation [31,36]. Whether yeasts can metabolize GRP and GR2P is unknown, but they have been shown to use glutathione and S-3-(hexan-1-ol)-glutathione, which is a precursor of varietal thiols as nitrogen sources [37,38].

Compounds of cluster I were characterized by their extremely low levels in Cinsault must and wines compared to other samples. This cluster was divided into three subclusters. Cluster I1 (51 features) contained phenolic compounds such as GRP, coutaric acid, catechin and procyanidin dimers, which were present in similar amounts in all Grenache and Syrah samples. Cluster I2—(49 features), overexpressed in Grenache musts and wines, contained the signals corresponding to caftaric and fertaric acid and some compounds containing nitrogen atoms that may be varietal markers, as proposed earlier [24].

Like those of cluster I2, compounds of cluster I3 (12 features) showed significantly higher concentrations in Grenache than in Syrah and were almost absent in Cinsault. However, all of them were more abundant in wines than in musts, meaning that they were formed in the course of fermentation. This was confirmed by the attribution of major signals to tryptophol, formed from tryptophan by yeast metabolism, and to coumarins and caffeic acid derivatives (e.g., caffeic ethyl ester) resulting from the degradation of grape phenolics.

Thus, the untargeted HRMS data highlighted markers related to the grape, which can either reflect varietal characteristics or be emphasized by vine-growing conditions and markers related to wine making. Among the former, anthocyanins appeared characteristic of Syrah, while hydroxycinnamic acids were particularly abundant in Grenache, as expected from earlier studies. Specific groups of nitrogen-containing compounds, including tryptophan in Grenache, and phenylalanine (i.e., the major precursor of phenolic compounds in plant biosynthesis) in Syrah were also detected as potential varietals markers, which is in agreement with earlier work [24]. These compounds were present in higher amounts in musts, indicating that they are metabolized or modified during fermentation. Consequently, tryptophan (i.e., tryptophol) and caffeic acid (e.g., ethyl caffeate) derivatives were more abundant in Grenache wines and can be considered as varietal markers revealed by fermentation. Similarly, norisoprenoid derivatives were identified as Cinsault markers formed during fermentation. Stilbenes appeared particularly low in Grenache samples, whereas flavonoid aglycones were more abundant in Cinsault samples. However, this might reflect an impact of vine-growing conditions rather than varietal characteristics, since solar radiation is known to impact the flavonol biosynthesis [39], whereas stilbenes take part in grape defense against fungi [30].

Caftaric acid, which is the major substrate of the grape polyphenoloxidase, and its glutathione derivatives formed after enzymatic oxidation (GRP, GRP2) were among the major compounds detected in Syrah and Grenache samples, but their concentration was extremely low in Cinsault samples. The hydroxycinnamic composition [40] as well as the hydroxycinnamic acid to glutathione ratio, which determines must sensitivity to enzymatic browning [5,15], are varietal characteristics. Similar amounts of hydroxycinnamic acids and glutathione have been reported for Syrah and Cinsault musts, whereas Grenache contained a higher level of caftaric and coutaric acids [5,15]. Nevertheless, the data obtained in the present study suggest that the Cinsault must was particularly oxidized compared to the other two. Comparison of the relative intensities of major hydroxycinnamic acid derivatives detected by HRMS with absorbance values determined by spectrophotometry indicates that most of the absorbance at 320 nm measured in the Cinsault samples is actually due to other compounds. These unknown compounds might correspond to the higher molecular weight compounds identified as markers of Cinsault musts and wines (cluster E2) and may be enzymatic oxidation products formed from hydroxycinnamic acids during must preparation. They might also correspond to the higher molecular weight compounds identified as markers of Cinsault musts and wines (cluster E2). Cinsault musts and wines also contained much lower amounts of all anthocyanins, although their pigment levels measured by spectrophotometry (A520) were intermediate between those of Syrah and Grenache samples. This indicates that most of the anthocyanins extracted from grape were converted to derived pigments in the early stages of the wine-making process. However, none of the expected derivatives (e.g., anthocyanin–caftaric acid adducts, pyranoanthocyanins) were detected in these samples, suggesting that if formed, they were rapidly involved into further reactions, which may also lead to higher molecular weight compounds.

The main characteristics of each cluster are summarized in Table 2.

Some differences were also observed between the two fermentation scales. A few features (15 in Syrah, 67 in Grenache, 18 in Cinsault) were significantly more abundant in the pilot-scale wine than in the corresponding laboratory-scale wine. Most of them were attributed to anthocyanins, flavonols and protocatechuic acid, along with caftaric acid in Grenache, suggesting a more important degradation of phenolic compounds at the laboratory scale, which was possibly due to increased oxidation. Accordingly, GRP showed a higher concentration in the Cinsault and Grenache LSW than in the PSW. However, flavan-3-ols which are known to be susceptible to oxidation were not affected, except for a procyanidin dimer, which was significantly lower in the Grenache LSW than in the PSW. Moreover, a number of N derivatives including tryptophan, phenylalanine and riboflavin were more abundant in the laboratory-scale wines, suggesting some differences in yeast metabolism.

## 3. Materials and Methods

### 3.1. Chemicals

HPLC-grade methanol was purchased from VWR Prolabo (Fontenay-sous-Bois, France). Ethanal, hydrochloric acid and potassium metabisulfite were purchased from Sigma Aldrich (St. Louis, MO, USA). Deionized water was obtained from a Milli-Q purification system (Millipore, Molsheim, France).

### 3.2. Must Preparation

Three different *Vitis vinifera* grape varieties, Cinsault, Grenache, and Syrah, were hand harvested at a potential degree of alcohol of 12° by the Centre du Rosé (IFV, Vidauban, France) in 2019. Overall, 300 kg of each grape variety was harvested in 20 kg bins in order to avoid grape crushing and uncontrolled maceration. Before the pre-fermentative process, grapes were kept at 12 °C for 12 h to be treated at the same temperature. Grapes were destemmed and crushed, and sodium hydrogen sulfite (2.5 g/100 kg) was added for protection against oxidation and microbial contamination. Musts were then put into in 3 hL tanks and submitted to controlled maceration at 17 °C during 4 h. After maceration, musts were added with sodium hydrogen sulfite (2.5 g/100 kg) to protect from oxidation and pressed. Then, musts were clarified by cold settling at 12 °C during 12 h before racking and adjusted at a turbidity of 135 NTU by adding solid particles collected during racking. Assimilable nitrogen content was adjusted at 110 mg/L for all musts using diammonium phosphate supplementation.

### 3.3. Fermentation

Prior to fermentation, each must has been homogenized using dry ice and distributed into three 100 L fermentation tanks (pilot scale), allowing to conduct each fermentation modality in triplicate. Homogenization was validated by colorimetric analysis, and analysis of variance was performed directly at the Centre du Rosé. A volume of 250 mL of must was collected from each of the nine fermentation tanks to fill nine 300 mL fermenters (laboratory scale). Fermenters and fermentation tanks were purged with CO_2_ pellets before filling and equipped with an airlock system to maintain anaerobiosis conditions. All fermenters were placed in a thermoregulated room set at 16 °C; they were placed on magnetic stirring (260 rpm) plates [41]. Active dry yeasts (K1 strain, ICV, Paris, France) were rehydrated for 20 min at 37 °C using a glucose solution in water (50 g/L) before inoculation at 0.4 g/L. Fermentation kinetics were manually monitored for 300 mL fermenters by weight losses induced by CO_2_ release. All wines were fermented to dryness (less than 2 g/L residual sugar) with no interruption for around 15 days. Color was monitored during pilot-scale fermentations, and 10 mL samples were collected for analysis at must and wine stage. Color analysis was performed directly, while composition analysis samples were stored at −20 °C until use.

### 3.4. Spectrophotometric Analysis

A Shimadzu UV-1900 UV–visible spectrophotometer was used to perform UV-Visible absorbance measurements following the protocols described by Atanasova et al., 2002 [42], using a 1 cm path length cell and adapting dilution to obtain absorbance values between 0.01 and 1.

Absorbance was measured directly on wines and musts from 230 to 800 nm. The three CIELAB color space parameters were calculated from the spectrum according to OIV method OIV-MA-AS2-11 [43] along with the hue angle (H = arctg$\frac{b*}{a*}$). Absorbance values at 520 nm (A_520_) were extracted from each spectrum to determine the colored fraction of pigments colored in wine conditions.

The total phenol index (TPI) at 280 nm, total red pigments (TRP) at 520 nm and absorbance at 320 nm (A_320_) were measured 20 min after the dilution of wines and musts in 1M HCl solution (TPI = A_280_ × F_D_ (dilution factor); TRP = A_520-HCl 1M_ × F_D_ and A_320_ = A_320-HCl 1M_ × F_D_).

Absorbance at 520 nm was measured 5 min after the addition of 60 μL of a sodium hydrogen sulfite solution (200 g/L) to 4 mL of wine or must to determine the sulfite bleaching resistant fraction of pigments (NBP = A_520-SO2_). A_520_ was also determined on musts and wines 40 min after the addition of 40 μL of a 12.6% (*v*/*v*) ethanal aqueous solution to 4 mL of samples to trap sulfites and convert bisulfite adducts to flavylium ions. This enabled calculation of the concentrations of bisulfite adducts (BA = A_520-ethanal_ − A_520_), flavylium forms (AH^+^ = A_520_ − A_520-SO2_) and hydrated forms (HF = TRP − A_520-ethanal_) of pigments susceptible to sulfite beaching.

### 3.5. Untargeted Analysis

#### 3.5.1. LC-HRMS Analysis

Analysis was operated on a UHPLC-DAD-HRMS system. Separations were performed using a Vanquish UHPLC-DAD (Thermo Fischer Scientific, San José, CA, USA) on a (10 × 1 mm i.d.) Acquity HSST3 column (Waters, Milford, MA, USA; 1.7 μm), operated at 35 °C. The mobile phase consisted of water/formic acid (99/1, *v*/*v*) (eluant A) and acetonitrile/water/formic acid (79.5/19.5/1, *v*/*v*) (eluant B). The flow rate was 0.22 mL/min. The elution program was as follows: isocratic for 1.5 min with 2% B, 2–12% B (1.5–4.5 min), isocratic with 12% B (4.5–7 min), 12–24% B (7–12 min), 24–48% B (12–15 min), 48–60% B (15–16 min), 60–100% B (16–17 min).

ESI-MS/MS analyses were performed with a Orbitrap Exploris 480 from Thermo Fisher Scientific (San José, CA, USA) equipped with an electrospray source and an internal post-source fluoranthene mass calibrant (radical ion at *m*/*z* 202.0777 in positive ion mode and *m*/*z* 202.0788 in negative ion mode). The spectrometer was operated in the positive ion mode (ion transfer tube: 280 °C; vaporizer temperature: 300 °C; sheath gaz, auxiliary gas and sweep gas: 40, 10 and 2 (arbitrary units), respectively; voltage set: 3.5 kV in positive mode, 2.5 kV in negative mode. The mass range was 150–1500 *m*/*z*.

The system performance was controlled by the injection of a standard mix (gallic acid, caffeic acid, (−)-epicatechin, rutin) at the beginning of each batch and compared to previous data. LockMass calibration was applied using the FlexMix™ calibration solution by direct infusion at 5 µL/min in both positive and negative ion mode.

The sample sequence was adapted from Arapitsas and Mattivi [11]. It includes wine and must samples, a Quality Control (QC) sample and a blank.

Samples were injected directly after centrifugation (15,000× *g*, 10 °C, 15 min) in biological triplicate following a randomized order. After equilibration, the sequence started with 3 injections of blank, which was followed by 3 injections of QC sample. Then, the QC sample and blank were injected with 6 real samples each. The sequence finished with 1 injection of QC sample and 1 injection of blank. The injection volume was set to 1 µL for all samples.

#### 3.5.2. Data Analysis

MS data from real samples, QC and blank samples were processed by Compound Discoverer software (v. 3.2.2.421, Thermo, Waltham, MA, USA) with a WorkFlow constituted by several steps: spectrum processing nodes which extract the mass spectral data from the input set file and align retention times of the compounds in the sequence; compound detection nodes which detect compounds by the extraction of masses as well as manage missing values, group adduct and isotopic peaks; and peak area refinement nodes which apply QC correction and remove background compounds found in the blank samples. Compound Discoverer parameters are available in the Appendix A. After Compound Discoverer data treatment, the number of features dropped from 1232 features to 755, each corresponding to a retention time associated with an *m*/*z* value. Features were then filtered based on the variability of the biological triplicate (CV < 10%) and the two-way ANoVA results (*p*-value < 0.05), keeping only features (729 features) with at least one modality exhibiting low variability and significant differences with another modality.

Annotation was performed manually by comparing retention times and mass spectra accuracy with a mass tolerance of 5 ppm based on the previous experience of the group with the specific instrumentation mass resolution and in accordance with the four levels of annotation described by Sumner [44] and presented in the Appendix A.

### 3.6. Statistical Analysis

A one-way analysis of variance on spectrophotometric data was performed with RStudio version 4.0.3 software (www.rstudio.com (accessed on 10 October 2020)) using a Student–Newman–Keuls post hoc test in the Agricolae package. ANOVA results were considered significant at *p*-value < 0.05. Untargeted composition analysis data (729 features dataset) were processed using Compound Discoverer version 3.1 software for two-way ANOVA, unsupervised hierarchical clustering and principal component analysis. Hierarchical clustering was generated using complete linkage and Euclidean distance methods and PCA using centered and scaled values.

## 4. Conclusions

In conclusion, a comparison of wines and musts made from Syrah, Grenache and Cinsault grapes using the same wine-making process highlighted important differences due to berry characteristics. Thus, musts and wines made from Syrah grapes showed higher absorbance values at 280 and 520 nm. Analysis of the HRMS data confirmed that Syrah samples (cluster A) contained higher concentrations of phenolic compounds and in particular of pigments such as anthocyanins and flavonols, reflecting the higher content of these molecules in Syrah grapes and/or their higher extraction rate. Grenache musts and wines were intermediate in terms of total phenol content but showed the highest values of A_320_. Accordingly, HRMS data revealed higher contents of caftaric acid derivatives in Grenache must (cluster G) and wines (cluster I2), which was consistent with the high concentrations of hydroxycinnamic acids reported earlier in Grenache grapes. It also enabled the detection of so far unreported caftaric oxidation products (caftaric–GRP adduct and GRP dimer) in musts. Tryptophan and tryptophol were also more abundant in Grenache samples. Cinsault musts and wines showed intermediate absorbance values at 280 and 520 nm but much lower levels of phenolic compounds (clusters I1, I2 and I3) than the Syrah and Grenache samples. This discrepancy indicates that absorbance in Cinsault samples is due to the presence of other compounds, which are poorly detected by HRMS and may be higher molecular weight compounds. Indeed, most of the higher molecular weight compounds detected were Cinsault markers, and they were more abundant in clusters E1, E2 and F. Future studies will aim at identifying these compounds as well as other markers of the grape variety and/or of the stage (before or after fermentation) detected by non-targeted HRMS analysis. Some differences were also exhibited between the two fermentation scales such as the abundance of nitrogen compounds, possibly derived from amino acids, suggesting some differences in yeast metabolism.

## Figures and Tables

**Figure 1 molecules-28-05748-f001:**
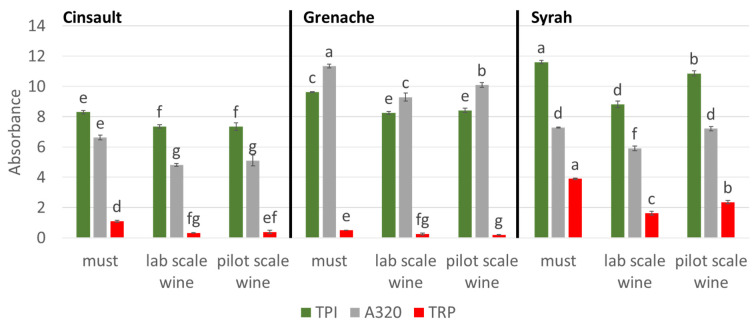
Total phenol index (TPI), absorbance values at 320 nm (A_320_) and total red pigments (TRP) in Grenache, Cinsault, and Syrah musts and wines fermented at laboratory and pilot scales; different superscripts indicate significant differences between samples (ANoVa with SNK test for *p*-value < 0.05).

**Figure 2 molecules-28-05748-f002:**
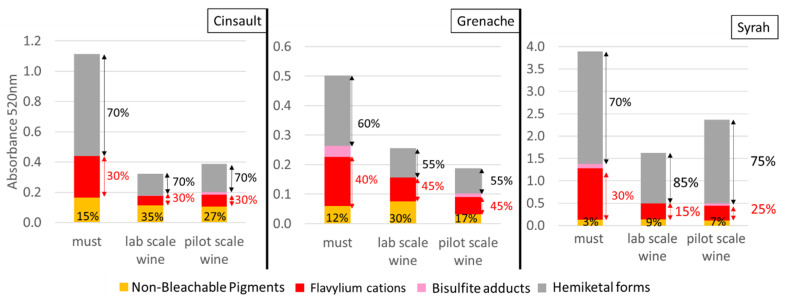
Absorbance at 520 nm under different conditions and contributions of the different forms of pigments in Syrah, Cinsault and Grenache musts and wines.

**Figure 3 molecules-28-05748-f003:**
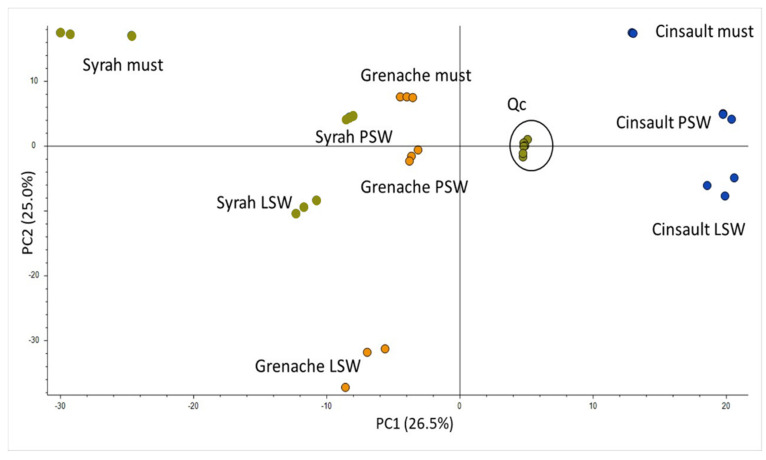
PCA plot of must and wine samples in the positive ion mode.

**Figure 4 molecules-28-05748-f004:**
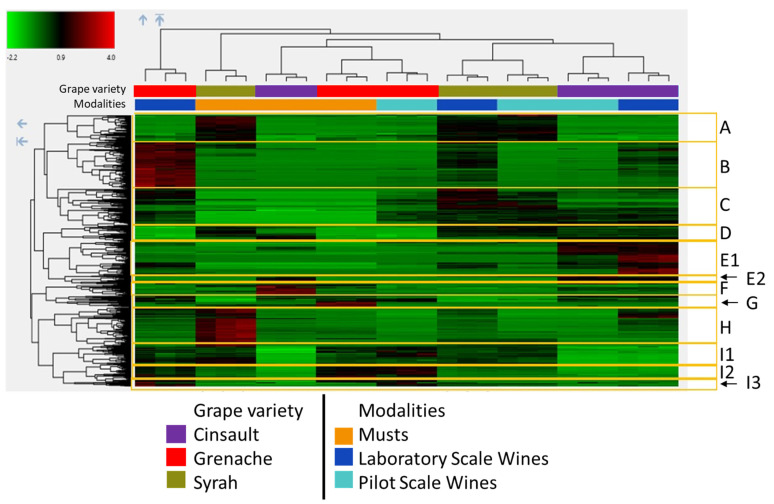
Unsupervised hierarchical clustering of metabolites and samples, showing the letter codes for metabolite clusters.

**Table 1 molecules-28-05748-t001:** CIELab parameters L* (lightness), a* (red/blue), and b* (yellow/green) and hue angle of must, lab-scale wine and pilot-scale wine made from the three grape varieties. Different superscript letters indicate significant differences between samples for a given parameter (ANoVa with SNK test for *p*-value < 0.05).

Grape Variety	Stage	L*	a*	b*	Hue
		Mean ± SD	Mean ± SD	Mean ± SD	Mean ± SD
Cinsault	must	75.10 ± 0.64 ^d^	16.29 ± 0.92 ^c^	17.49 ± 1.06 ^a^	0.82 ± 0.06 ^c^
Cinsault	lab-scale wine (LSW)	88.82 ± 0.59 ^b^	6.05 ± 0.51 ^f^	13.20 ± 0.64 ^b^	1.14 ± 0.05 ^a^
Cinsault	pilot-scale wine (PSW)	89.29 ± 0.53 ^b^	8.18 ± 1.14 ^e^	15.72 ± 1.22 ^ab^	1.09 ± 0.07 ^a^
Grenache	must	87.43 ± 0.90 ^b^	11.25 ± 0.52 ^d^	14.01 ± 0.50 ^b^	0.89 ± 0.01 ^c^
Grenache	lab-scale wine (LSW)	89.83 ± 2.90 ^b^	4.87 ± 0.24 ^f^	10.40 ± 2.89 ^c^	1.12 ± 0.10 ^b^
Grenache	pilot-scale wine (PSW)	93.57 ± 0.19 ^a^	4.94 ± 0.26 ^f^	6.69 ± 0.48 ^de^	0.97 ± 0.02 ^c^
Syrah	must	53.48 ± 2.83 ^e^	47.65 ± 1.16 ^a^	8.52 ± 0.56 ^cd^	0.18 ± 0.01 ^e^
Syrah	lab-scale wine (LSW)	75.87 ± 1.81 ^d^	28.55 ± 0.47 ^b^	7.06 ± 1.46 ^de^	0.24 ± 0.05 ^d^
Syrah	pilot-scale wine (PSW)	78.98 ± 0.74 ^c^	28.71 ± 0.58 ^b^	4.97 ± 0.40 ^e^	0.17 ± 0.01 ^d^

**Table 2 molecules-28-05748-t002:** Main characteristics of each cluster.

Cluster (Number of Features)	Compounds	Variety	Process Stage	Marker of/Origin
A (73)	Anthocyanins	Syrah		Variety
B (126)	Unknown N compounds	Higher in Grenache	Lab-scale wines	Fermentation
C (103)	Yeast metabolites		wines	Fermentation
D (43)	Stilbenes Rearrangement product of mv3glc?	Lower in Grenache		Fungal pressure in the vineyard?
E1 (93)	Norisoprenoids flavonoids N compounds higher Mw N and S compounds	Higher in Cinsault	Higher in wine	Variety, Environmental conditions revealed by fermentation
E2 (24)	Higher molecular weight compounds	Higher in Cinsault	Higher in wine	Reactions in wine
F (32)	N compounds	Cin > Gre > Syr	Higher in musts	Grape
G (16)	Tryptophan derivatives Caftaric–GRP adducts, GRP dimer	Higher in Grenache	Higher in musts	Grape variety/vine growing Grape variety PPO oxidation Degradation in wine
H (107)	N compounds (Phe) peonidin 3-glucoside flavonol derivatives GRP isomer, GRP2	Syrah	Higher in musts	Variety, Anthocyanin profile GSH/AH ratio PPO oxidation Degradation in wine
I1 (51)	GRP, coutaric acid, flavan-3-ols	Lower in Cinsault		Variety
I2 (49)	Caftaric, fertaric N compounds	Lower in Cinsault Higher in Grenache		Variety
I3 (12)	Tryptophol Ethyl caffeic coumarin	Lower in Cinsault Higher in grenache	Higher in wine	Fermentation

## Data Availability

The data presented in this study are available in the Appendix A and on https://zenodo.org/ (accessed on 26 July 2023).

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
