# Peer review of "Untargeted Metabolomics Approach Using UHPLC-HRMS to Unravel the Impact of Fermentation on Color and Phenolic Composition of Rosé Wines"

_molecules, 2023, doi:10.3390/molecules28155748_

Round 1

Reviewer 1 Report

Table 1 is illegible. It should be explained somewhere what the parameters a, b and L are. You have to read the whole paragraph of text below the table to understand it.

Line 460 80,5 + 19,5 + 1,0 = 101%

Material and methods must come before "Results and Discussion". In the current layout, the work is completely illegible.

Lines 233-237: This paragraph should be expanded. The description is too vague. It lacks important information.

Sample preparation was omitted. This has to be completed in detail, because the way the sample is prepared affects the results obtained and the results of the statistical analysis.

Injection volume not given. Was the injection volume always the same or the same for specific groups of samples (Musts, PSW, LSW)?

In the abstract, the authors write: “This study focuses on changes occurring during alcoholic fermentation of Syrah, Grenache and Cinsault musts, conducted at laboratory (250mL) and pilot (100 L) scales”. Albeit I am unable to find a single word about this in the conclusions - what changes were detected by the HR MS technique?

Conclusion is unacceptable. The entire paragraph is devoted to UV-Vis analyses, and only the last sentence states that the compounds may be identified in the future HR MS. In other words, there is nothing in the Conclusions about the obtained HR MS results, their interpretation and conclusions from their interpretation. The most important thing is also missing - a clear formulation of the scientific novelty. What have the authors discovered? How have they improved the state of the art in this field?

The text is written in understandable language. The content is linguistically easy to understand.

Author Response

  • Table 1 is illegible. It should be explained somewhere what the parameters a, b and L are. You have to read the whole paragraph of text below the table to understand it.

Specification were added from line 108 and in the Table 1 title.

  • Line 460 80,5 + 19,5 + 1,0 = 101%

Correction was made (79,5 + 19,5 + 1,0=100).

  • Material and methods must come before "Results and Discussion". In the current layout, the work is completely illegible.

This layout follows instructions for author provided by Molecules (https://www.mdpi.com/journal/molecules/instructions) but we agree with your comment.

  • Lines 233-237: This paragraph should be expanded. The description is too vague. It lacks important information.Sample preparation was omitted. This has to be completed in detail, because the way the sample is prepared affects the results obtained and the results of the statistical analysis.

The samples were injected directly after centrifugation (Information added line 534).

  • Injection volume not given. Was the injection volume always the same or the same for specific groups of samples (Musts, PSW, LSW)? 

Information added line 538

  • In the abstract, the authors write: “This study focuses on changes occurring during alcoholic fermentation of Syrah, Grenache and Cinsault musts, conducted at laboratory (250mL) and pilot (100 L) scales”. Albeit I am unable to find a single word about this in the conclusions - what changes were detected by the HR MS technique? Conclusion is unacceptable. The entire paragraph is devoted to UV-Vis analyses, and only the last sentence states that the compounds may be identified in the future HR MS. In other words, there is nothing in the Conclusions about the obtained HR MS results, their interpretation and conclusions from their interpretation. The most important thing is also missing - a clear formulation of the scientific novelty. What have the authors discovered? How have they improved the state of the art in this field?

HRMS data were already described in the conclusion which mentioned specificities of the different clusters visible on the heat map représentation of this data. However, we modified some sentences so as to highlight HRMS results

Reviewer 2 Report

In this manuscript the authors investigated the impact of fermentation on color and phenolic composition of Rosé wines made from Syrah, Grenache and Cinsault grapes, using spectrophotometric method and untargeted metabolomics based on ultra-high performance liquid chromatography coupled to high resolution mass spectrometry.

This is an important topic as color is a major quality trait of rosé wines.

My main concern with the authors' discussion is the accuracy of the untargeted metabolomic approach using UHPLC-HRMS. As stated by the author, future studies will aim at identifying these compounds detected by non-targeted HRMS analysis. However, the accuracy of non-targeted HRMS analysis is related to the reliability of the conclusion of this manuscript. Thus,

I suggest adding a discussion on the accuracy of the method (non-targeted HRMS analysis), or adding the quantitative determination of phenolic composition of Rosé wines.

Other issues should be addressed:

1.     Line 34 to line 39: cite appropriate references.

2.     Line 66, (2h, 8h): Put a blank between numbers and units.

3.     Line 77 (SEC): For abbreviations that appear for the first time, the full name must be noted.

4.     Line 435 (by Atanasova et al.): different font styles.

5.     Line 477 (1.1.1 Data analysis), 491 (1.2 Statistical analysis): sections were wrongly numbered.

Author Response

In this manuscript the authors investigated the impact of fermentation on color and phenolic composition of Rosé wines made from Syrah, Grenache and Cinsault grapes, using spectrophotometric method and untargeted metabolomics based on ultra-high performance liquid chromatography coupled to high resolution mass spectrometry.

This is an important topic as color is a major quality trait of rosé wines.

  • My main concern with the authors' discussion is the accuracy of the untargeted metabolomic approach using UHPLC-HRMS. As stated by the author, future studies will aim at identifying these compounds detected by non-targeted HRMS analysis. However, the accuracy of non-targeted HRMS analysis is related to the reliability of the conclusion of this manuscript. Thus, I suggest adding a discussion on the accuracy of the method (non-targeted HRMS analysis), or adding the quantitative determination of phenolic composition of Rosé wines.

Information about the accuracy of the method were added from line 528 to 531. Unfortunetly, quantitative method were not available at the lab during the acquisition of the data. 

  • Other issues should be addressed:
  1. Line 34 to line 39: cite appropriate references.

References have been added

  1. Line 66, (2h, 8h): Put a blank between numbers and units.

modification done

  1. Line 77 (SEC): For abbreviations that appear for the first time, the full name must be noted.

modification done

  1. Line 435 (by Atanasova et al.): different font styles.

This has been corrected.

  1. Line 477 (1.1.1 Data analysis), 491 (1.2 Statistical analysis): sections were wrongly numbered.

This has been corrected.

Reviewer 3 Report

The manuscript presents an interesting contribution to reach a better understanding on how the lab and pilot scale winemaking can differ. It also brings new information to research in rose wines. 

Some additions, especially regarding the type of data analysis will improve the quality of the manuscript. 

Suggestions: 

Materials and Methods

-          Specify the type of ANOVA, one-way ANOVA or two-way ANOVA and include reasoning why.

-          Include information about the heatmap. Was it performed on raw data? Or on the dimensions obtained from the PCA? If so, on how many dimensions.  

Results

Add abbreviations LSW and PSW to Table 1

I would suggest moving the figures to a part of the text where the information contained has already started to be described. Also, in Figure 2, add the full name of the variables analysed.

Line 210, you can add a citation of works which have analysed colour and phenolic composition during fermentation in some of these cultivars, and how your results compare to those findings.

Lines 240-243 How is each variable contributing to each dimension. Grenache PSW or Syrah PSW are probably not being influenced by PC1 neither PC2 but by other dimensions.

The use of PCA is a widely used and valid option to analyse the data. However, it will be worth to include a brief reasoning on why you choose to perform unsupervised strategy. What are the advantages and limitations to your results compared to other possibilities in chemometrics which have been applied to oenology.

For example, and since you are evaluating two different matrices (unfermented and fermented), why not to follow a supervised approach since you are evaluating two different matrices (unfermented and fermented samples) using approaches such as PCA-DA or OPLS-DA. There has been work published on Shiraz wine phenolics using LC-HRMS and both data strategies, unsupervised and supervised. Ultimately, data could have been treated as a multi block type of dataset considering must, pilot scale and lab scale as different block datasets. 

For figure 4, it would facilitate the reading if the coloring of cultivar and sampling stage are not the same. Also, add the scaling.

Add specifications about parameters for the heatmap.

-          Did you filter compounds with a low variability?

-          What criteria do you follow to define the number of clusters. How do you interpret what is a cluster and what subclusters?

-          What type of ANOVA was performed for clusters.

A criterion could to define that from A to E2 is cluster 1 and from F to I3 is cluster 2 and the rest is within cluster variability. Similarly, the clusters for the different treatments could be only 2, 3 or 4 clusters.  

In addition to specify the type of ANOVA performed on the clusters, I suggest adding an extra column to Table 2 where it specifies what variable is significant (cultivar, modality or the interaction between both).

Author Response

The manuscript presents an interesting contribution to reach a better understanding on how the lab and pilot scale winemaking can differ. It also brings new information to research in rose wines.

Some additions, especially regarding the type of data analysis will improve the quality of the manuscript.

Suggestions:

Materials and Methods

-          Specify the type of ANOVA, one-way ANOVA or two-way ANOVA and include reasoning why.

One-way ANOVA because grape varieties were selected for their color differences that were already known as different.

-          Include information about the heatmap. Was it performed on raw data? Or on the dimensions obtained from the PCA? If so, on how many dimensions. 

It was performed on data obtained after treatment, a specification was added in the text (L251 and 547-552).

Results

Add abbreviations LSW and PSW to Table 1

modification done

I would suggest moving the figures to a part of the text where the information contained has already started to be described. Also, in Figure 2, add the full name of the variables analysed.

Figures have been moved as suggested and requested information added to the text.

Line 210, you can add a citation of works which have analysed colour and phenolic composition during fermentation in some of these cultivars, and how your results compare to those findings.

Modification were added to the text.

Lines 240-243 How is each variable contributing to each dimension. Grenache PSW or Syrah PSW are probably not being influenced by PC1 neither PC2 but by other dimensions.

The use of PCA is a widely used and valid option to analyse the data. However, it will be worth to include a brief reasoning on why you choose to perform unsupervised strategy. What are the advantages and limitations to your results compared to other possibilities in chemometrics which have been applied to oenology.

For example, and since you are evaluating two different matrices (unfermented and fermented), why not to follow a supervised approach since you are evaluating two different matrices (unfermented and fermented samples) using approaches such as PCA-DA or OPLS-DA. There has been work published on Shiraz wine phenolics using LC-HRMS and both data strategies, unsupervised and supervised. Ultimately, data could have been treated as a multi block type of dataset considering must, pilot scale and lab scale as different block datasets.

 The data was first treated using unsupervised Principal Component Analysis (PCA) to avoid any bias in sample discrimination resulting from overfitting of supervised methods.

We are aware that supervised analysis such as PLS is commonly used to maximize separation between classes of samples and highlight features discriminating them. However, such approaches are prone to over-fitting and cross-validation is required to avoid this bias. Unsupervised analysis (e.g. PCA or Hierarchical clustering) is more reliable and should generally be preferred provided it discriminates samples according to the variation sources investigated. In our case, unsupervised methods proved sufficiently discriminant and thus efficient to highlight markers of varieties and modalities when applied on our data set.

We also agree that we could have treated the data using multi-block analysis and thank the reviewer for this suggestion.

 For figure 4, it would facilitate the reading if the coloring of cultivar and sampling stage are not the same. Also, add the scaling.

Figure 4 has been modified as requested.

Add specifications about parameters for the heatmap. (l533-536) done

-          Did you filter compounds with a low variability?

Yes, we did: only features with at least one modality exhibiting low variability (CV< 10%) and significant differences (ac (p-value<0.05, cording to two-way ANoVA) with another modality were kept. Description of feature selection has been added lines 554-559 

-          What criteria do you follow to define the number of clusters. How do you interpret what is a cluster and what subclusters?

Clusters were defined as 3 nodes on the hierarchical classification and 4 nodes as sub-clusters, enabling discrimination of modalities (variety or must, LSW, PSW). Added lines 506-508

-          What type of ANOVA was performed for clusters. Two-way ANOVA (l 279, 518 & 520)

A criterion could to define that from A to E2 is cluster 1 and from F to I3 is cluster 2 and the rest is within cluster variability. Similarly, the clusters for the different treatments could be only 2, 3 or 4 clusters.

Yes, it could have been done as said but the different modalities (variety, wine-making stage) were better discriminated from the third node of the hierarchical clustering.

In addition to specify the type of ANOVA performed on the clusters, I suggest adding an extra column to Table 2 where it

Informations will be added in the supplementary dataset and the table 2 has also been modified following your suggestions.

Round 2

Reviewer 2 Report

The paper can be considered to be published in this journal

Reviewer 3 Report

The current version of the manuscript can be considered for publication